# Radio Frequency Cavity's Analytical Model and Control Design

Mahsa Keikha * , Jalal Taheri Kahnamouei and Mehrdad Moallem

School of Mechatronic Systems Engineering, Simon Fraser University, Surrey, BC V3T 0A3, Canada; jtaherik@sfu.ca (J.T.K.); mmoallem@sfu.ca (M.M.)
* Correspondence: mkeikha@sfu.ca

**Abstract:** Reduction or suppression of microphonic interference in radio frequency (RF) cavities, such as those used in Electron Linear Accelerators, is necessary to precisely control accelerating fields. In this paper, we investigate modeling the cavity as a cylindrical shell and present its free vibration analysis along with an appropriate control scheme to suppress vibrations. To this end, we first obtain an analytical mechanical dynamic model of a nine-cell cavity using a modified Fourier-Ritz method that provides a unified solution for cylindrical shell systems with general boundary conditions. The model is then verified using the ANSYS software in terms of a comparison of eigenfrequencies which prove to be identical to the proposed model. We also present an active observer-based vibration control scheme to suppress the dominant mechanical modes of the cavity. The control system performance is investigated using simulations.

**Keywords:** radio frequency superconducting cavity; electron linear accelerator; flexural dynamic; microphonic noise cancellation; kalman filter; dynamic modeling; control designing

## 1. Introduction

In electron linear accelerators (e-LINACs), electrons are accelerated up to 50MeV along a linear beam line. Multi-cell superconducting Radio-Frequency (RF) cavities, such as the nine-cell niobium cavity of Advanced Rare IsotopE Laboratory (ARIEL) accelerator at TRIUMF, Canada's particle accelerator center, accelerate charged particles via an oscillating electric field known as the accelerating field [1,2].

To deliver a high-quality beam requires that the phase of the accelerated particles be precisely controlled so that bunched particles receive the same amount of energy from the multi-cell RF cavities. However, these cavities are subject to impact by microphonic interference. This interference, created primarily by environmental mechanical vibrations, can cause deformations in the shape of the cavity that create a shift in resonance frequency [3]. In attempts to assure good field stability through a well-tuned cavity, various studies to suppress mechanical vibrations have been conducted in accelerator labs around the world [4–6].

An accurate model of the system is required to design a controller for suppressing microphonic interference. Analytical solutions for RF fields in an RF structure are only available in simple geometries. Creating an analytical model of mechanical vibrations in a multi-cell cavity is an extremely complex task. For instance, there are restrictions in measuring deflection variables of a multi-cell niobium cavity since access to the cavity is restricted to either end since it is suspended within a Helium bath. A structure without such limitations would allow for placement of sensors and actuators on arbitrary locations cavities; however, the cavity mechanism limits application of force only to the cavity ends.

There exist several approaches for vibration analysis of cylindrical shells such as the Rayleigh-Ritz method [7–13]. Active noise cancellation in cylindrical shells has also been worked out extensively, e.g. [14–24]. Utilization of piezoelectric laminated cylindrical shells for active vibration control was studied by several studies, e.g. [25–27]. However, these studies assume that one can place actuators at arbitrary locations on the shell.

This paper presents a unified solution for cylindrical shells systems with generic boundary conditions using a modified Fourier-Ritz approach. Each displacement for the cylindrical shell is expressed as the modified Fourier series plus auxiliary functions, regardless of the boundary and continuity constraints. The Rayleigh-Ritz method is used to calculate all expansion coefficients as generalized coordinates. A major challenge is choosing the best actuator locations that can be used to actively cancel dominant cylindrical shell's natural frequencies. For a case where access is limited to the two ends of the cylindrical shell, the only choice available is to apply horizontal forces (e.g., using piezoelectric stacks) at both ends of the cylinder, which we investigate in this paper. To this end, we provide a modeling scheme for approximating a nine-cell cavity in Section 2. In Section 3 we introduce an observer-based LQG controller which is a combination of a Kalman filter and LQR controller. Sections 4 and 5 present the results in terms of accuracy and effectiveness through MATLAB and simulation analysis. Conclusions are presented in Section 6.

## 2. Development a 3D Cylindrical Shell Equivalent Model of a Nine-cell RF Cavity

In this section, we develop a cylindrical shell model using apply shell theory and Rayleigh-Ritz method. To this end we obtain the shell's kinetic and potential energy, displacement functions, and mode shape equations. In the cylindrical shell, the whole energy function consists mainly of two components: potential energy $E_{Potential}$ and kinetic energy $E_{Kinetic}$. As part of the Rayleigh-Ritz method, the energy function is used to formulate the equations of motion. Next, we determine the cylindrical shell's equation of motion derived from the Lagrangian equation, and its stiffness matrix and mass matrix. Using a modified Fourier series to satisfy the boundary conditions, we arrive at an expanded equation of motion. This improved Fourier series is composed of a standard Fourier series and auxiliary polynomial functions. The displacements of the cylindrical shell component $(u, v, w)$ can be written with the consideration of the symmetric modes. Next, assuming that the actuator forces can be applied only at both ends, and using the method of virtual work, we determine the input matrix.

### 2.1. Description of Nine-Cell RF Cavity

A multi-cell cavity is a structure with multiple resonators (cells) coupled together as shown in Figure 1. The effective length of the nine-cell Cavity is $L = 1.061$ with a wall thickness of $h = 3$ μm. The cavity is fabricated from solid niobium sheets.

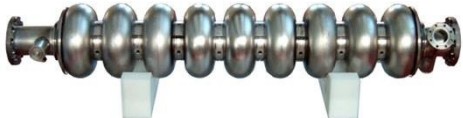

**Figure 1.** TRIUMF's Nine-cell Cavity Actual Structure.

In Figure 2, side view and geometry of TRIUMF's nine-cell cavity is shown. A helium tank contains the superfluid helium needed for cooling. It also serves as a mechanical support of the cavity and as a part of the tuning mechanism.

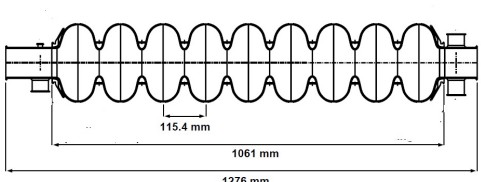

**Figure 2.** Nine-cell Cavity Side View.

### 2.2. Description of the Cylindrical Shell Model

In this study, we use a simplified equivalent model of the cavity and a cylindrical coordinate system of $(x, \theta, r)$ for a cylindrical shell which has length $L$, thickness $h$, and radius $R$. The displacement functions for this cylindrical shell are $u, v, w$ in $x$, $\theta$ and $r$ directions. The thickness of the shell is assumed to be uniform and very small, compared to the length of the cylindrical shell [28]. Hence one can apply shell theory for structural modeling.

### 2.3. Kinetic and Potential Energy for Cylindrical Shell

Using the classical theory of shells for a circular cylindrical shell, the general displacements of the cylindrical shell with respect to the Figure 3 coordinate system are denoted by $v, u, w$ in the $x$, $\theta$ and $r$ directions, respectively (see Appendix A).

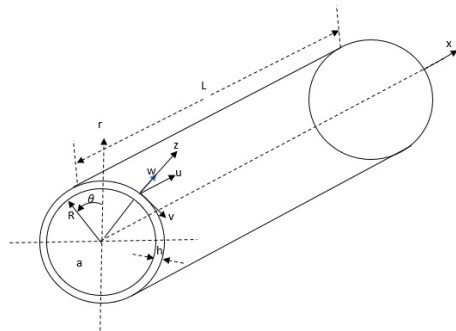

**Figure 3.** Cylindrical Shell Coordinate System.

To find a solution for the equation of motion, Mirsky [29] suggested use of displacement function potentials as $\Phi$ and $\Psi$ functions (See Appendix A (A4), (A5), (A6)). Considering the boundary condition for the cylindrical shell with a finite length and thin wall, according to Reissner's thin shell theory [30], the displacement functions are as follows

$$u(x, \theta, t) = U \cos(\lambda x) \cos(n\theta) e^{j\omega t} \tag{1}$$

$$v(x, \theta, t) = V \sin(\lambda x) \sin(n\theta) e^{j\omega t} \tag{2}$$

$$w(x, \theta, t) = W \sin(\lambda x) \cos(n\theta) e^{j\omega t} \tag{3}$$

where $U$, $V$ and $W$ are constants. The displacement functions can be expressed as the series of functions in $x$ and $\theta$ directions that have $m$ longitudinal and $n$ transverse nodes, respectively (see Appendix A). The kinetic energy $E_{Kinetic}$ and the potential energy $E_{Potential}$ of a cylindrical shell are provided in Appendix B.

#### 2.3.1. Rayleigh-Ritz Method

The Rayleigh-Ritz method is a direct numerical method of approximating eigenvalues considering boundary conditions. The method relies on approximating the shell's structural deformation. It enables one to reduce an infinite number of degrees-of-freedom (DOF) of a system into a finite number. Using the Rayleigh–Ritz method, we next derive a dynamic model for the free vibration analysis of a cylindrical shell.

$$u(x, \theta, t) = \sum_{n=1}^{\infty} u_n(x, \theta, t) = \sum_{n=1}^{\infty} \Phi_u(x) \cos n\theta q_{un}(t) \tag{4}$$

$$v(x, \theta, t) = \sum_{n=1}^{\infty} v_n(x, \theta, t) = \sum_{n=1}^{\infty} \Phi_v(x) \sin n\theta q_{vn}(t) \tag{5}$$

$$v(x, \theta, t) = \sum_{n=1}^{\infty} w_n(x, \theta, t) = \sum_{n=1}^{\infty} \Phi_w(x) \cos n\theta q_{wn}(t) \tag{6}$$

The terms $\Phi_u(x)$, $\Phi_v(x)$ and $\Phi_w(x)$ are mode shape function vectors in $u$, $v$ and $w$ directions, respectively, which satisfy the boundary conditions.

$$\Phi_u(x) = [\cos\frac{\pi x}{L}, \cos\frac{2\pi x}{L}, \cos\frac{3\pi x}{L}, ..., \cos\frac{m\pi x}{L}] \tag{7}$$

$$\Phi_v(x) = [\sin\frac{\pi x}{L}, \sin\frac{2\pi x}{L}, \sin\frac{3\pi x}{L}, ..., \sin\frac{m\pi x}{L}] \tag{8}$$

$$\Phi_w(x) = [\sin\frac{\pi x}{L}, \sin\frac{2\pi x}{L}, \sin\frac{3\pi x}{L}, ..., \sin\frac{m\pi x}{L}] \tag{9}$$

The terms $\delta_{un}$, $\delta_{vn}$ and $\delta_{wn}$ are generalized displacement vector of each direction for transverse $n$-th mode.

$$\delta_{un}(t) = [\delta_{un1}, \delta_{un2}, \delta_{un3}, ..., \delta_{unm}] \tag{10}$$

$$\delta_{vn}(t) = [\delta_{vn1}, \delta_{vn2}, \delta_{vn3}, ..., \delta_{vnm}] \tag{11}$$

$$\delta_{wn}(t) = [\delta_{wn1}, \delta_{wn2}, \delta_{wn3}, ..., \delta_{wnm}] \tag{12}$$

The kinetic energy for the cylindrical shell considering the Rayleigh-Ritz method is given by

$$E_{Kinetic} = \frac{1}{2}\rho h L \pi (\dot{\delta}_u^T M_{uu}\dot{\delta}_u + \dot{\delta}_v^T M_{vv}\dot{\delta}_v + \dot{\delta}_w^T M_{ww}\dot{\delta}_w). \tag{13}$$

In the kinetic energy equation, the components are defined to be elements of the mass matrix M, denoted by $M_{uu}$, $M_{vv}$ and $M_{ww}$ in the u and v and w directions, respectively, as follows

$$M_{uu} = \int_0^L \Phi_u^T \Phi_u dx = \Phi_{uu} \tag{14}$$

$$M_{vv} = \int_0^L \Phi_v^T \Phi_v dx = \Phi_{vv} \tag{15}$$

$$M_{ww} = \int_0^L \Phi_w^T \Phi_w dx = \Phi_{ww} \tag{16}$$

and the potential energy with the same method is

$$E_{Potential} = \frac{ERh\pi}{2L(1-\mu^2)}(\delta_u^T K_{uu}\delta_u + \delta_v^T K_{vv}\delta_v + \delta_w^T K_{ww}\delta_w +$$
$$2\delta_u^T K_{uv}\delta_v + 2\delta_v^T K_{vw}\delta_w + 2\delta_u^T K_{uw}\delta_w). \tag{17}$$

In the potential energy equation, the terms of K, or stiffness matrix, are denoted by $K_{uu}$, $K_{vv}$ and $K_{ww}$ in the $u$ and $v$ and $w$ directions, respectively. Also, $K_{uv}$, $K_{vw}$ and $K_{uw}$ are given by

$$K_{uu} = \frac{(1-\mu^2)L^2n^2}{2R^2}\int_0^L \Phi_u^T\Phi_u dx + \int_0^L \dot{\Phi}_u^T\dot{\Phi}_u dx = \frac{(1-\mu^2)L^2n^2}{2R^2}\Phi_{uu} + \dot{\Phi}_{uu} \tag{18}$$

$$K_{vv} = (\frac{L^2n^2}{R^2})\int_0^L \Phi_v^T\Phi_v dx + \frac{(1-\mu)}{2}\int_0^L \dot{\Phi}_v^T\dot{\Phi}_v dx = (\frac{L^2n^2}{R^2})\Phi_{vv} + \frac{(1-\mu)}{2}\dot{\Phi}_{vv} \tag{19}$$

$$K_{ww} = (\frac{L^2}{R^2} + \frac{L^2n^4h^2}{12R^4})\int_0^L \Phi_w^T\Phi_w dx + \frac{(1-\mu^2)h^2n^2}{6R^2}\int_0^L \dot{\Phi}_w^T\dot{\Phi}_w dx +$$
$$\frac{h^2}{12L^2}\int_0^L \ddot{\Phi}_w^T\ddot{\Phi}_w dx - \frac{\mu^2h^2n^2}{6R^2}\int_0^L \ddot{\Phi}_w^T\Phi_w dx \tag{20}$$

$$K_{uv} = \frac{(1 - \mu^2)Ln}{2R} \int_0^L \dot{\Phi}_u^T \dot{\Phi}_v dx + \frac{\mu Ln}{R} \int_0^L \dot{\Phi}_u^T \Phi_v dx \qquad (21)$$

$$K_{vw} = \frac{L^2 n}{R^2} \int_0^L \Phi_v^T \Phi_w dx \qquad (22)$$

$$K_{uw} = \frac{\mu L}{R} \int_0^L \Phi_u^T \dot{\Phi}_w dx. \qquad (23)$$

### 2.3.2. Equation of Motion for the Cylindrical Shell Structure

From the energy method, using Rayleigh-Ritz equations, and considering the general boundary conditions, the equation of motion for cylindrical shell can be derived from Lagrange's equation, i.e.,

$$LAG = E_{Potential} - E_{Kinetic}. \qquad (24)$$

According to Hamiltonian's principle, the variation of the proceeding function is set to zero with respect to expansion of coefficients

$$\frac{d}{dt}\left(\frac{\partial LAG}{\partial \dot{\delta}}\right) - \frac{\partial LAG}{\partial \delta} = 0. \qquad (25)$$

Inserting the displacement equations into the Lagrangian equation and minimizing it against all the unknown coefficients, a system of linear algebraic equation in matrix form can be obtained as

$$(\rho RLh\pi)M\ddot{\delta} + \frac{Rh\pi E}{(1 - \mu)^2 L}K\delta = 0 \qquad (26)$$

$K$ is the cylindrical shell's stiffness matrix and $M$ is the cylindrical shell's mass matrix. (See details in Appendix C).

$$K = \begin{bmatrix} K_{uu} & K_{uv} & K_{uw} \\ K_{uv}^T & K_{vv} & K_{vw} \\ K_{uw}^T & K_{vw}^T & K_{ww} \end{bmatrix} \qquad (27)$$

$$M = \begin{bmatrix} M_{uu} & 0 & 0 \\ 0 & M_{vv} & 0 \\ 0 & 0 & M_{ww} \end{bmatrix} \qquad (28)$$

The natural frequencies and eigenvectors of the cylindrical shell can be derived by solving a standard eigenvalue problem. Each of the eigenvectors contains a Fourier coefficient for that corresponding mode

$$|K - \Omega^2 M| = 0 \qquad (29)$$

where $\Omega = L\sqrt{\frac{\rho(1-\mu^2)}{E}}\omega$.

The selection of the displacement auxiliary functions is very important when analyzing the vibration characteristics of the cylindrical shell for high accuracy and convergent results. Researchers have investigated the free vibration of thin walled cylindrical shells under different displacement auxiliary functions (see e.g. [10]).

### 2.3.3. Modified Fourier Series

In the Rayleigh-Ritz energy method, the auxiliary functions are the essential to achieving an accurate solution. These auxiliary functions need to satisfy the boundary conditions. In this study, using the modified Fourier series method, we write the displacement functions in $u$ and $v$ directions as

$$u(x,\theta,t) = e^{j\omega t}\left[\sum_{n=0}^{\infty}\sum_{m=0}^{\infty}U_{nm}\cos(\lambda_m x)\cos(n\theta)+\right.$$
$$\sum_{n=0}^{\infty}\sum_{p=1}^{2}\hat{U}_{np}\alpha_p(x)\cos(n\theta)+$$
$$\sum_{n=1}^{\infty}\sum_{m=0}^{\infty}\tilde{U}_{nm}\cos(\lambda_m x)\sin(n\theta)+$$
$$\left.\sum_{n=1}^{\infty}\sum_{p=1}^{2}\hat{\tilde{U}}_{np}\alpha_p(x)\sin(n\theta)\right] \tag{30}$$

$$v(x,\theta,t) = e^{j\omega t}\left[\sum_{n=0}^{\infty}\sum_{m=0}^{\infty}V_{nm}\cos(\lambda_m x)\cos(n\theta)+\right.$$
$$\sum_{n=0}^{\infty}\sum_{p=1}^{2}\hat{V}_{np}\alpha_p(x)\cos(n\theta)+$$
$$\sum_{n=1}^{\infty}\sum_{m=0}^{\infty}\tilde{V}_{nm}\cos(\lambda_m x)\sin(n\theta)+$$
$$\left.\sum_{n=1}^{\infty}\sum_{p=1}^{2}\hat{\tilde{V}}_{np}\alpha_p(x)\sin(n\theta)\right]. \tag{31}$$

The auxiliary functions used in displacement functions in the u and v direction of the cylindrical shell equations are

$$\alpha_1(x) = \frac{x}{L^2}(x-L)^2 \tag{32}$$

$$\alpha_2(x) = \frac{x^2}{L^2}(x-L) \tag{33}$$

and the displacement functions in $w$ directions is

$$w(x,\theta,t) = e^{j\omega t}\left[\sum_{n=0}^{\infty}\sum_{m=0}^{\infty}W_{nm}\cos(\lambda_m x)\cos(n\theta)+\right.$$
$$\sum_{n=0}^{\infty}\sum_{f=1}^{4}\hat{W}_{nf}\beta_f(x)\cos(n\theta)+$$
$$\sum_{n=1}^{\infty}\sum_{m=0}^{\infty}\tilde{W}_{nm}\cos(\lambda_m x)\sin(n\theta)+$$
$$\left.\sum_{n=1}^{\infty}\sum_{f=1}^{4}\hat{\tilde{W}}_{nf}\beta_f(x)\sin(n\theta)\right]. \tag{34}$$

The auxiliary functions used in displacement functions in the $w$ direction

$$\beta_1(x) = \frac{L}{12\pi}\left(27\sin(\frac{\pi x}{2L}) - \sin(\frac{3\pi x}{2L})\right) \tag{35}$$

$$\beta_2(x) = -\frac{L}{12\pi}\left(27\cos(\frac{\pi x}{2L}) + \cos(\frac{3\pi x}{2L})\right) \tag{36}$$

$$\beta_3(x) = \frac{L^3}{3\pi^3}\left(3\sin(\frac{\pi x}{2L}) - \sin(\frac{3\pi x}{2L})\right) \tag{37}$$

$$\beta_4(x) = -\frac{L^3}{3\pi^3}\left(3\cos(\frac{\pi x}{2L}) - \cos(\frac{3\pi x}{2L})\right) \tag{38}$$

where $L$ is the length of the cylindrical shell.

All these polynomial auxiliary functions satisfy the boundary conditions at $x = 0, L$. According to the equation of motion previously discussed, we have $(K - \omega^2 M)D = 0$, where $D$ is the coefficient vector in $u$, $v$ and $w$ directions

$$D = [U, V, W]^T. \tag{39}$$

So the equation of motion can be expressed as follows

$$(K - \omega^2 M) \begin{bmatrix} U \\ V \\ W \end{bmatrix} = \begin{bmatrix} 0 \\ 0 \\ 0 \end{bmatrix} \tag{40}$$

where the vector in the $u$ direction is

$$
\begin{aligned}
U = [&U_{00}, U_{01}, U_{02}, ..., U_{nm}, ..., U_{NM} \\
&\hat{U}_{00}, \hat{U}_{01}, \hat{U}_{02}, ..., \hat{U}_{nm}, ..., \hat{U}_{NM} \\
&\tilde{U}_{00}, \tilde{U}_{01}, \tilde{U}_{02}, ..., \tilde{U}_{np}, ..., \tilde{U}_{NP} \\
&\hat{U}_{00}, \hat{U}_{02}, ..., \hat{U}_{np}, ..., \hat{U}_{NP}]^T
\end{aligned} \tag{41}
$$

and the vector in the $v$ direction is

$$
\begin{aligned}
V = [&V_{00}, V_{01}, V_{02}, ..., V_{nm}, ..., V_{NM} \\
&\hat{V}_{00}, \hat{V}_{01}, \hat{V}_{02}, ..., \hat{V}_{np}, ..., \hat{V}_{NP} \\
&\tilde{V}_{00}, \tilde{V}_{01}, \tilde{V}_{02}, ..., \tilde{V}_{nm}, ..., \tilde{V}_{NM} \\
&\hat{V}_{00}, \hat{V}_{02}, ..., \hat{V}_{np}, ..., \hat{V}_{NP}]^T.
\end{aligned} \tag{42}
$$

Furthermore, the vector in $w$ direction is given by

$$
\begin{aligned}
W = [&W_{00}, W_{01}, W_{02}, ..., W_{nm}, ..., W_{NM} \\
&\hat{W}_{00}, \hat{W}_{01}, \hat{W}_{02}, ..., \hat{W}_{nf}, ..., \hat{W}_{NF} \\
&\tilde{W}_{00}, \tilde{W}_{01}, \tilde{W}_{02}, ..., \tilde{W}_{nm}, ..., \tilde{W}_{NM} \\
&\hat{\tilde{W}}_{00}, \hat{\tilde{W}}_{02}, ..., \hat{\tilde{W}}_{nf}, ..., \hat{\tilde{W}}_{NF}]^T
\end{aligned} \tag{43}
$$

where $M$ and $N$ are the truncated for $m$ (longitudinal modes) and $n$ (transverse modes), respectively. Now if a force is applied to the cylindrical shell the virtual work is given by

$$\delta W_{virtual} = E_{Kinetic} - E_{Potential} \tag{44}$$

and the equation of motion is

$$M^* \ddot{\delta} + K^* \delta = B^* u(t) \tag{45}$$

where $u(t)$ is input force to the cylindrical shell and B is the force participation matrix, which reflects the effect of applied force on each mode on the cylindrical shell. Each column in $B$ matrix represent a set of force from an actuator.

$$B^* = \begin{bmatrix} B_u \\ B_v \\ B_w \end{bmatrix}_{4N(M+P)+2N(M+F) \times n_{input}} \tag{46}$$

where our defined input $B$ matrix in the $u$ direction is

$$B_u = [B_{u00}, B_{u01}, B_{u02}, ..., B_{unm}, ..., B_{uNM}$$
$$\tilde{B}_{u00}, \tilde{B}_{u01}, \tilde{B}_{u02}, ..., \tilde{B}_{unp}, ..., \tilde{B}_{uNP}$$
$$\hat{B}_{u00}, \hat{B}_{u01}, \hat{B}_{u02}, ..., \hat{B}_{unm}, ..., \hat{B}_{uNM}$$
$$\hat{\hat{B}}_{u00}, \hat{\hat{B}}_{u02}, ..., \hat{\hat{B}}_{unp}, ..., \hat{\hat{B}}_{uNP}]^T_{(2N(M+P)) \times n_{input}}. \tag{47}$$

The input matrix $B$ in the $v$ direction is given by

$$B_v = [B_{v00}, B_{v01}, B_{v02}, ..., B_{vnm}, ..., B_{vNM}$$
$$\tilde{B}_{v00}, \tilde{B}_{v01}, \tilde{B}_{v02}, ..., \tilde{B}_{vnp}, ..., \tilde{B}_{vNP}$$
$$\hat{B}_{v00}, \hat{B}_{v01}, \hat{B}_{v02}, ..., \hat{B}_{vnm}, ..., \hat{B}_{vNM}$$
$$\hat{\hat{B}}_{v00}, \hat{\hat{B}}_{v02}, ..., \hat{\hat{B}}_{vnp}, ..., \hat{\hat{B}}_{vNP}]^T_{(2N(M+P)) \times n_{input}} \tag{48}$$

and lastly, the input matrix $B$ in the $w$ direction is

$$B_w = [B_{w00}, B_{w01}, B_{w02}, ..., B_{wnm}, ..., B_{wNM}$$
$$\tilde{B}_{w00}, \tilde{B}_{w01}, \tilde{B}_{w02}, ..., \tilde{B}_{wnp}, ..., \tilde{B}_{wNP}$$
$$\hat{B}_{w00}, \hat{B}_{w01}, \hat{B}_{w02}, ..., \hat{B}_{wnm}, ..., \hat{B}_{wNM}$$
$$\hat{\hat{B}}_{w00}, \hat{\hat{B}}_{w02}, ..., \hat{\hat{B}}_{wnp}, ..., \hat{\hat{B}}_{wNP}]^T_{(2N(M+F)) \times n_{input}}. \tag{49}$$

Depending on the force direction and position, and the effects of force on vibration modes, the $B$ matrix can be determined. As can be seen, the resulting displacements in $u$, $v$ and $w$ are a combination of sine and cosine modes.

## 3. Control Design

The equation of motion for (45) contains infinite number of modes of vibration. Our goal in control design is just to control a limited number of those modes, considering the fact that we only have access to both ends of the cylinder. By applying forces at both ends, we want to cancel out some natural modes of vibration. Next we study the model for controllability of modes for a reduced order model.

By solving the eigenvalue problem corresponding to $n$ modes, the eigenvalues and eigenvectors are obtained. The mass and stiffness matrices need to satisfy the orthogonality condition. Thus we have

$$O_n^T M^* O_n = I \tag{50}$$

$$O_n^T K^* O_n = \Lambda_n \tag{51}$$

where $I$ is identity matrix, $O_n$ is eigenvector matrix, and $\Lambda_n$ is eigenvalue matrix for $n$ eigenvalues. Furthermore, $\Lambda_n$ is given by

$$\Lambda_n = \begin{bmatrix} \omega_1^2 & & & & & \\ & \omega_1^2 & & & & \\ & & \omega_2^2 & & & \\ & & & \omega_2^2 & & \\ & & & & \ddots & \\ & & & & & \omega_n^2 \end{bmatrix}. \tag{52}$$

After rearranging the natural frequencies in ascending order, $N$ vibration modes are considered. The equation of motion with $N$ natural frequencies, and taking into consideration the damping coefficient, is then given by

$$\ddot{\hat{\delta}} + \Gamma_N \dot{\hat{\delta}} + \Lambda_N \hat{\delta} = O_N^T B^* u. \tag{53}$$

In this study, we define state-space control model

$$\dot{\Delta}(t) = A\Delta(t) + Bu(t) + w(t) \tag{54}$$

$$y(t) = C\Delta(t) + Du(t) + v(t) \tag{55}$$

where the $A$ matrix is given by

$$A = \begin{bmatrix} 0 & I \\ -\Lambda_N & -\Gamma_N \end{bmatrix}. \tag{56}$$

The components of $A$ matrix are given by

$$\Gamma_N = 2Z\Omega_N = \begin{bmatrix} 2\zeta_1\omega_1 & & & & \\ & \ddots & & & \\ & & 2\zeta_j\omega_j & & \\ & & & \ddots & \\ & & & & \zeta_N\omega_N \end{bmatrix} \tag{57}$$

where $A$ is the system matrix, $B$ is the input matrix, and $C$ is the output matrix. The term $w(t)$ is the external disturbance (if there is any) and $v(t)$ is the sensor noise. Also we have $C = B^T$ where

$$B = \begin{bmatrix} 0 \\ O_N^T B_N^* \end{bmatrix} \tag{58}$$

and

$$C = \begin{bmatrix} C_N O_N & 0 \end{bmatrix}. \tag{59}$$

*3.1. Reduced-Order Modeling for State-Space Matrices*

A reduced order state-space model of the system is given by

$$\dot{\Delta}_r(t) = A_r \Delta_r(t) + B_r u_r(t) + w(t) \tag{60}$$

$$y_r(t) = C_r \Delta_r(t) + D_r u_r(t) + v(t) \tag{61}$$

where

$$A_r = \begin{bmatrix} 0 & I \\ -\Omega_r & -\Gamma_r \end{bmatrix} \tag{62}$$

where each components of $A_r$ matrix are defined as

$$\Omega_r = diag(\omega_i^2) \quad n = 0, 1, 2, ..., r \tag{63}$$

and

$$\Gamma_r = diag(2\zeta_i\omega_i) \quad n = 0, 1, 2, ..., r \tag{64}$$

where $\omega_j$ is the frequency of mode j, $\zeta_j$ is the effective modal damping of mode j, and $u_r$ is the vector of input forces

$$u_r = \begin{bmatrix} u_1(t) \\ \vdots \\ u_r(t) \end{bmatrix} \tag{65}$$

$B_r$ is a ($2n \times n$ input) state-space matrix defined by where n input is the number of scalar input forces.

$$B_r = \begin{bmatrix} 0 \\ \Phi_r \end{bmatrix} \tag{66}$$

where

$$\Phi_r = O_r^T F_u \tag{67}$$

where $O_r$ is the matrix of reduced order eigenvectors and $F_u$ is a unit force matrix with size (ndof $\times$ ninput). It has 1 at the degrees of freedom where input forces are active and 0 elsewhere. Now that the states $\delta$ have been expressed as a function of the input loads, the equation for the degrees of freedom observed (outputs $y_r$) is written as:

$$y_r = \begin{bmatrix} y_r(t) \\ \dot{y}_r(t) \\ \ddot{y}_r(t) \end{bmatrix} = C_r \Delta_r + D_r F \tag{68}$$

$C_r$ is a (3*noutput $\times$ 2*$n$) state-space matrix, where noutput is derived from outputs.

$$C_r = \begin{bmatrix} \Psi_r & 0 \\ 0 & \Psi_r \\ -\Psi_r \Omega_r & -\Psi_r \Gamma_r \end{bmatrix} \tag{69}$$

where

$$\Psi_r = U_u O_r \tag{70}$$

$U_u$ is a unit displacement matrix with size (noutput $\times$ ndof). It has 1 on degrees of freedom where output is requested and 0 elsewhere. $D_r$ is a (3*noutput $\times$ ninput) state-space matrix defined by

$$D_r = \begin{bmatrix} 0 \\ 0 \\ \Psi_r \Phi_r \end{bmatrix}. \tag{71}$$

### 3.2. Controllability of the Reduced Order Model

The system in (54) is controllable if mass and stiffness matrices ($M^*$ and $K^*$) are symmetric, diagonal and positive definite, and we assume that we only have access to both ends of the model and the boundary condition is two both ends are free and our model is fixed in the midway. The system controllability matrix is given by

$$Q_C = \begin{bmatrix} B_r & A_r B_r & A_r^2 B_r & A_r^3 B_r & ... & A_r^{2n-1} B_r \end{bmatrix}. \tag{72}$$

To evaluate the controllability of the system, we should prove that the controllability matrix is full rank, i.e., $rank(Q_C)= 2n$. In order to calculate the rank of $Q_C$, we obtain columns of the controllability matrix as follows

$$\begin{cases} A_r^{2i} = (-\Omega_r)^i I_{2n} & i = 0, 2, 4, ..., n \\ A_r^{2i-1} = (-\Omega_r)^i A_r & i = 1, 3, 5, ..., n-1. \end{cases} \tag{73}$$

Considering (73), in general, we are unable to demonstrate that $Q_C$ is full rank, meaning that the system is not fully controllable (for all of its natural frequencies). In order to develop a controller, we typically first divide the system into controllable and non-controllable matrices. The controller is then designed to suppress vibrations in the controllable modes. However, for this specific dynamic and special boundary condition that we assume the mass and stiffness matrices as diagonal matrix, we can prove that $Q_C$ is full rank and the system is controllable. The controllability Gramian of the system is defined as

$$W_C(t) = \int_0^t e^{\tau A_r^T} B_r B_r^T e^{\tau A} d\tau. \tag{74}$$

To prove the controllability of the system, considering the reduced order system, and taking all the mentioned assumptions into consideration, we see that the controllability matrix is not full rank or rank($Q_C$) $\neq 2n$. Here we need to decompose the controllable and non-controllable modes and then design the controller for the modes that are controllable.

Controllable and Un-Controllable Decomposition

Kalman decomposition in control theory [31] presents a mathematical method for converting a model of any linear time-invariant ($LTI$) control system to a form in which the system may be decomposed into a standard form that clearly shows the system's observable and controllable components. Considering the state space model of the reduced order system in (54)

$$\hat{\Delta}_r = \begin{bmatrix} \Delta_{CO} \\ \Delta_{C\hat{O}} \\ \Delta_{\hat{C}O} \\ \Delta_{\hat{C}\hat{O}} \end{bmatrix} \tag{75}$$

$$\hat{A}_r = \begin{bmatrix} A_{CO} & 0 & A_{13} & 0 \\ A_{12} & A_{C\hat{O}} & A_{23} & A_{24} \\ 0 & 0 & A_{\hat{C}O} & 0 \\ 0 & 0 & A43 & A_{\hat{C}\hat{O}} \end{bmatrix} \tag{76}$$

$$\hat{B}_r = \begin{bmatrix} B_{CO} \\ B_{C\hat{O}} \\ 0 \\ 0 \end{bmatrix} \tag{77}$$

$$\hat{C}_r = \begin{bmatrix} C_{CO} & 0 & C_{\hat{C}} & 0 \end{bmatrix} \tag{78}$$

$$\hat{D}_r = D_r. \tag{79}$$

Therefore, our new model for the control design will be the controllable and observable part of our finite element analysis model.

$$\dot{\Delta}_{CO} = A_{CO}\Delta_{CO} + B_{CO}u \tag{80}$$

$$y = C_{CO}\Delta_{CO} + D_{CO}u \tag{81}$$

where $\Delta_{CO}$ and $A_{CO}$ matrix and $B_{CO}$ matrix are

$$\Delta_{CO} = \begin{bmatrix} \delta_{CO} \\ \dot{\delta}_{CO} \end{bmatrix}. \tag{82}$$

Controllability Gramian of the new system is defined as

$$\hat{W}_C(t) = \int_0^t e^{\tau A C O^T} B_{CO} B_{CO}^T e^{\tau A_{Co}} d\tau = \int_0^t e^{\tau A_{CO}^T} e^{\tau A_{CO}} d\tau. \tag{83}$$

The system is controllable if, and only if, $\hat{W}_C$ is nonsingularr for ant $t > 0$. Also, controllability matrix for this new defined system is

$$\hat{Q}_C = \begin{bmatrix} B_{CO} & A_{CO}B_{CO} & A_{CO}^2 B_{CO} & A_{CO}^3 B_{CO} & \dots & A_{CO}^{2n-1} B_{CO} \end{bmatrix} \tag{84}$$

$rank(\hat{Q}_C) = 2n$ and it is full rank. Therefore, the new defined system is controllable.

### 3.3. Lyapunov-Based Controller

The system introduced in (80), is asymptotically stable using the following controller

$$u = -G\Delta \tag{85}$$

where $G$ is a positive gain and $\dot{\delta}$ is replaced with an Observer based control design as follows

$$\dot{\hat{\Delta}} = A_{\hat{\Delta}}\hat{\Delta} + B_{CO}u \tag{86}$$

where $A_{\tilde{\Delta}}$ ($2n \times 2n$) matrix is

$$A_{\tilde{\Delta}} = \begin{bmatrix} l_1 \times I & I \\ -\Omega_{CO} + l_2 \times I & 0 \end{bmatrix} \tag{87}$$

the input to the observer-based controller is

$$u = -G\hat{\Delta}. \tag{88}$$

Defining the Lyapunov function of the system as $V_L$, the derivative of this specified Lyapunov function candidate ($\dot{V}_L$) should be negative definite or negative semi-definite, thus

$$V_L = \frac{1}{2}\dot{\Delta}^T M^* \dot{\Delta} + \frac{1}{2}\Delta^T K^* \Delta \tag{89}$$

$$\dot{V}_L = \dot{\Delta}^T M^* \ddot{\Delta} + \Delta^T K^* \dot{\Delta} = \dot{\Delta}^T B^* u. \tag{90}$$

If the input is defined as multiplication of a negative value and input $B^*$ matrix, then the derivative of the Lyapunov function is

$$\dot{V}_L = -G\dot{\Delta}^T \dot{\Delta} \leq 0. \tag{91}$$

Therefore, the first derivative of the Lyapunov function is negative semi definite and the system is asymptotically stable.

### 3.4. Observability

In this section, we show the system observability through a lemma, then we proceed to design an observer-based controller. Observability Gramian for our system is defined as

$$\hat{W}_O(t) = \int_0^t e^{\tau A_{CO}^T} C_{CO}^T C_{CO} e^{\tau A_{CO}} d\tau. \tag{92}$$

System is observable if and only if $W_0(t)$ is nonsingular for any $t > 0$. Observability matrix of this system is given by

$$\hat{Q}_O = \begin{bmatrix} C_{CO} & C_{CO}A_{CO} & C_{CO}A_{CO}^2 & \dots & C_{CO}A_{CO}^{2n-1} \end{bmatrix}^T. \tag{93}$$

To prove the observability of the system we should have $rank(\hat{Q}_O) = 2n$ that is full rank.

### 3.5. Observer-Based Control Design

The design of an observer to estimate the states of the system in order to design a controller is as follows

$$\dot{\hat{\Delta}} = A_{CO}\hat{\Delta} + B_{CO}u + R(y - \hat{y}). \tag{94}$$

Recalling the system output y from (81), the estimated output of the system is defined as

$$\hat{y} = C_{CO}\hat{\Delta} + Du \tag{95}$$

putting (95) into (94) and simplifying our observer equation becomes

$$\dot{\hat{\Delta}} = (A_{CO} - RC_{CO})\hat{\Delta} + B_{CO}u. \tag{96}$$

The estimation error of the states is written as

$$\tilde{\delta} = \hat{\delta} - \delta_{CO}. \tag{97}$$

Therefore, the observer error system can be defined as

$$\dot{\tilde{\Delta}} = \dot{\hat{\Delta}} - \dot{\Delta_{CO}} = (A_{CO} - RC_{CO})\tilde{\Delta} \tag{98}$$

where $R$ is an observer gain $(2n \times 1)$ vector.

$$\tilde{A} = A_{CO} - RC_{CO} \tag{99}$$

Since the system is observable, we can arbitrarily place eigenvalues of (96) in the left half plane.

### 3.6. Observer-Based Controller Transfer Function

Considering (81), let us define input as follows

$$u = -G\Delta_{CO} \tag{100}$$

where $u$ is input to the controller and the observer gain vector is

$$\dot{\hat{\Delta}} = A_{CO}\hat{\Delta} + B_{CO}(-G\hat{\delta}) + RC_{CO}\hat{\Delta} - Ry. \tag{101}$$

Our observer-based controller system can be written in a matrix form as follows

$$\dot{\hat{\Delta}} = A_{\hat{\Delta}}\hat{\Delta} - Ry \tag{102}$$

$$u = -G\hat{\Delta} \tag{103}$$

where

$$A_{\hat{\Delta}} = A_{CO} - B_{CO}G - RC_{CO}. \tag{104}$$

The transfer function of the observer-based controller system is

$$H_{\Delta}(s) = \frac{U(s)}{Y(s)} = -G \times (SI - A_{\hat{\Delta}})^{-1} \times R. \tag{105}$$

To calculate the transfer function of this observer-based controller we need to find the determinant of $A_{\hat{\Delta}}$ matrix. This determinant is greater than zero, therefore, $A_{\hat{\Delta}}$ is invertible and the transfer function can be calculated. See Figure 4 that shows the block diagram of observer-based controller.

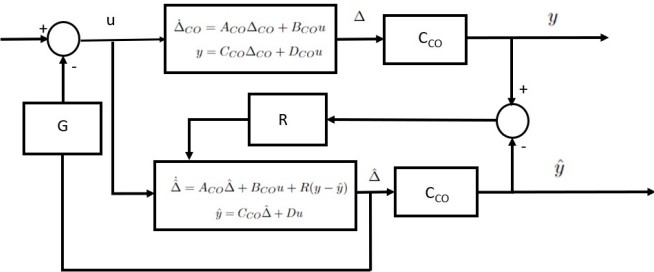

**Figure 4.** Observer Control Model.

### 3.7. LQG Control Design

A Kalman filter [32] has been designed to estimate the system states from the sensor measurement. The dynamic equation of an optimal observer is

$$\dot{\hat{\Delta}} = A_{CO}\hat{\Delta} + B_{CO}u_K + R_K(y - \hat{y}) \tag{106}$$

$$y = C_{CO}\hat{\Delta} \tag{107}$$

where $R_K = PC_{CO}^T T^{-1}$ , in which P is positive definite solution of the algebraic Riccati equation

$$PA_{CO}^T + A_{CO}P - PC_{CO}^T T^{-1}C_{CO}P + J = 0. \tag{108}$$

Based on Linear Quadratic Regulator (LQR) theory, $J \geq 0$ and $T \ggg 0$, the control input is $u_K = -G_K\hat{x}(t)$. And $G_K = T^{-1}B_{CO}^T S$ that S is the positive definite solution of the algebraic Riccati equation

$$SA_{CO}^T + A_{CO}S - SB_{CO}^T T^{-1}B_{CO}S + J = 0. \tag{109}$$

Actual states of the system $\delta(t)$ are not available so $\hat{\delta}(t)$ that are new observed states from the sensors are used for calculating the control force.

$$u_K = -G_K\hat{\Delta}(t) \tag{110}$$

and the equation that combines the observer and the controller is

$$\dot{\hat{\Delta}} = A_{CO}\hat{\Delta}(t) - B_{CO}G_K\hat{\Delta}(t) + R_K(y_r - C_{CO}\hat{\Delta}) = (A_{CO} - B_{CO}G_K - R_KC_{CO})\hat{\Delta}(t) + R_Ky \tag{111}$$

A proper control input is based on the measured signal of the collocated piezo sensor and the LQG controller has been developed in MATLAB code (see Figure 5).

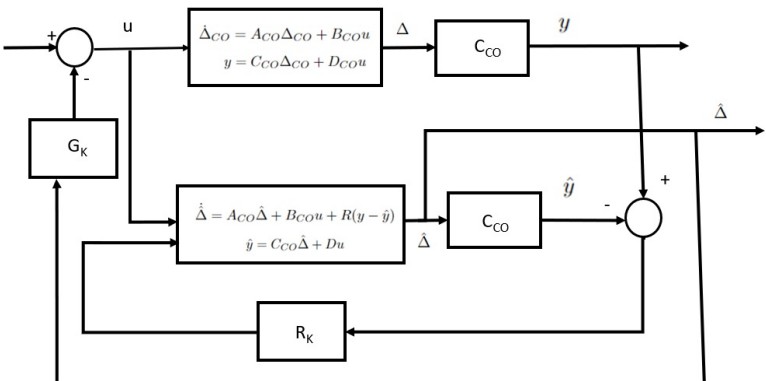

**Figure 5.** Kalman Filter Observer-Control Model.

## 4. Comparison of Numerical Model with ANSYS Results

In this study, the goal is vibration control in a nine-cell superconducting cavity. To study the cavity structure and vibration analysis, the model has been simplified as shown in Figure 6. The closest structure to the simplified nine-cell cavity is a cylindrical shell with the length $L = 1.062$ m and thickness of $h = 2.8$ mm (See Figure 7). ANSYS modeling of the cylindrical shell with the same characteristics for the nine-cell cavity is the same vibration and modal analysis as the analytical solution with modified Fourier model.

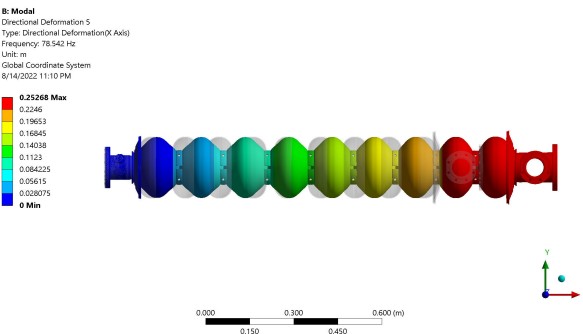

**Figure 6.** ANSYS Simulation for Simplified Cavity.

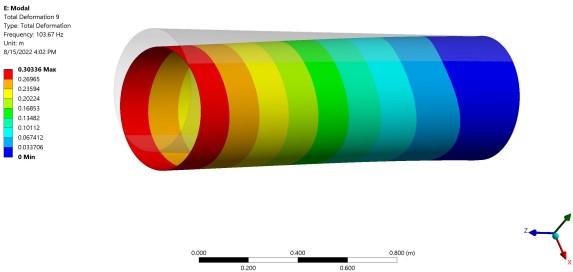

**Figure 7.** ANSYS Simulation for Cylindrical Shell Mode Shapes.

## 5. Simulation ANALYSIS and MATLAB RESULTS

The boundary conditions are that we only have access to the both ends of the cavity. By applying pairs of forces we want to cancel out some modes of vibration. Considering our constraints, we apply two equal forces to the both ends in the opposite direction of each other. Simulation analysis is carried out through 24 points selected by 4 points for longitudinal direction and 6 points for circumferential direction as shown in Figure 8. In the first attempt of the simulation, the first force has been applied to the node 1, and the second force to the node 23.

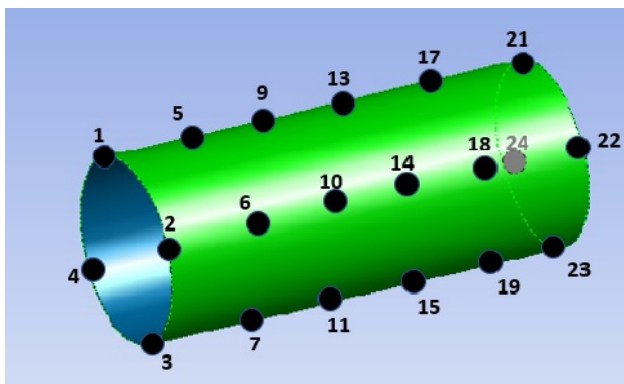

**Figure 8.** Simulation Setup for Modal Analysis.

In order to control more modes of vibration, we increase the quantity of the paired forces that are applying to both ends. In this case, by applying 4 pairs of forces to the both ends nodes (1, 2, 3, 4) on one side and nodes (21, 22, 23, 24) on the other side, the rank of observability and controllability matrix become 8. It means that, according to our proposed model, when 4 pairs of forces are applied to both ends, then 8 modes of vibration can be controlled. Again, we decompose the system into controllable and uncontrollable parts. Also, to design the observer we use the observable part of the matrix from the decomposition. In this case, the rank of observability and controllability of the system are both 4. We decompose the system into controllable and uncontrollable parts and design

a controller, based on the designed observer for those 4 controllable modes of vibrations, with an observer-based controller and with a LQG controller. To continue our studies we design an observer base controller with pole placement in the LHP (left half plane).

In Figure 9 the inputs for the system (which is the input voltage to the actuators) have been shown.

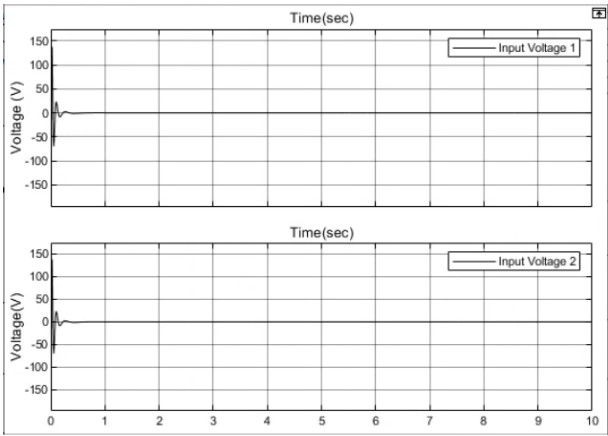

**Figure 9.** System's Inputs in MATLAB.

Figure 10 is showing the output of the system which are sensors voltages.

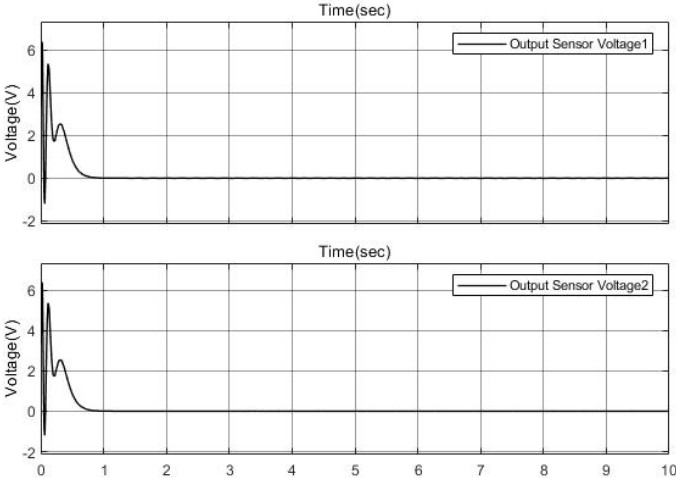

**Figure 10.** System's Output.

As it can be seen in Figure 11, the observer error is zero.

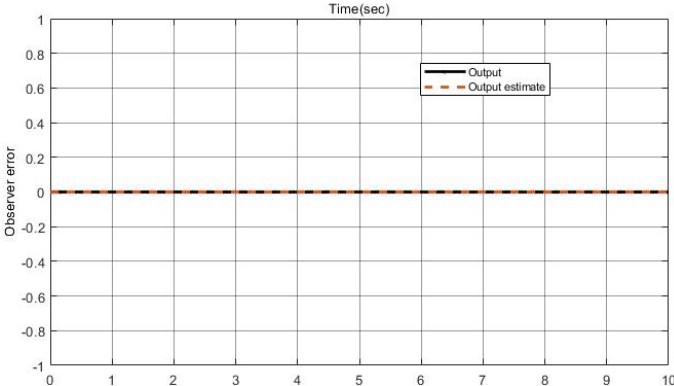

**Figure 11.** Observer Error.

Figure 12 illustrates that the designed observer-based controller is perfectly canceling out the vibration for those four modes.

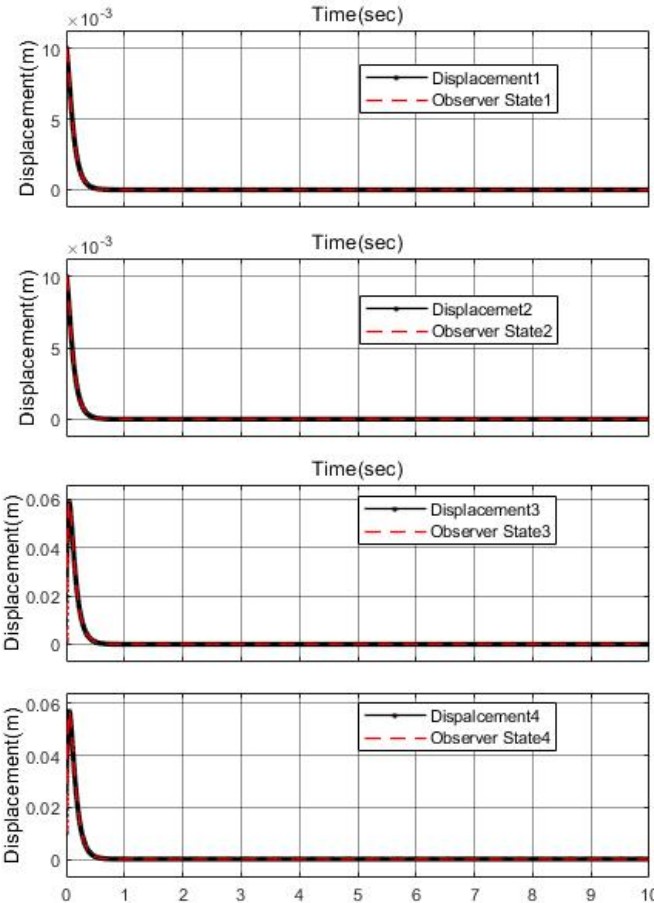

**Figure 12.** System and Observer with Four Controlled Modes in MATLAB.

In Figure 13 the states of the system without control and after applying the controller is shown. As can be seen in this figure, the observer-based LQG controller is controlling the 8 modes of vibration.

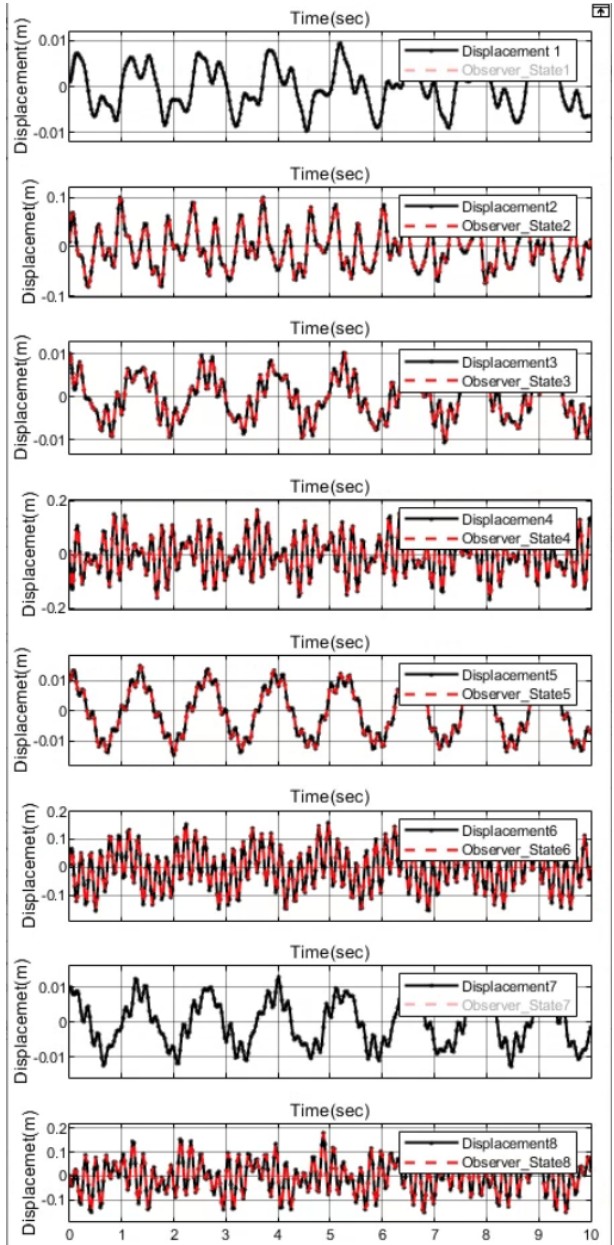

**Figure 13.** No Control and Controlled System States Signals for 8 modes in MATLAB.

In Figure 14, the output of the system with and without controller has been shown.

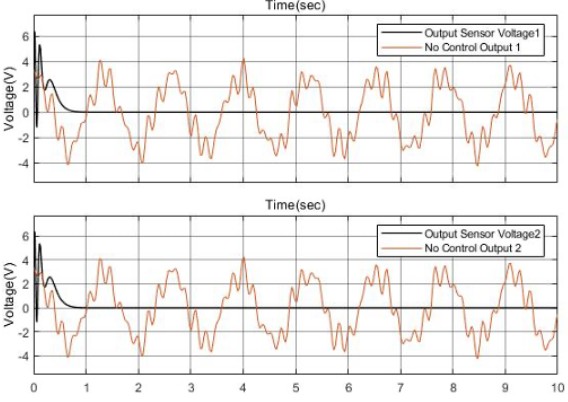

**Figure 14.** System's Output Signals in MATLAB.

Figure 15 it is shown that the observer-based LQG controller is controlling the 8 modes and cancelling out the unwanted vibration from them.

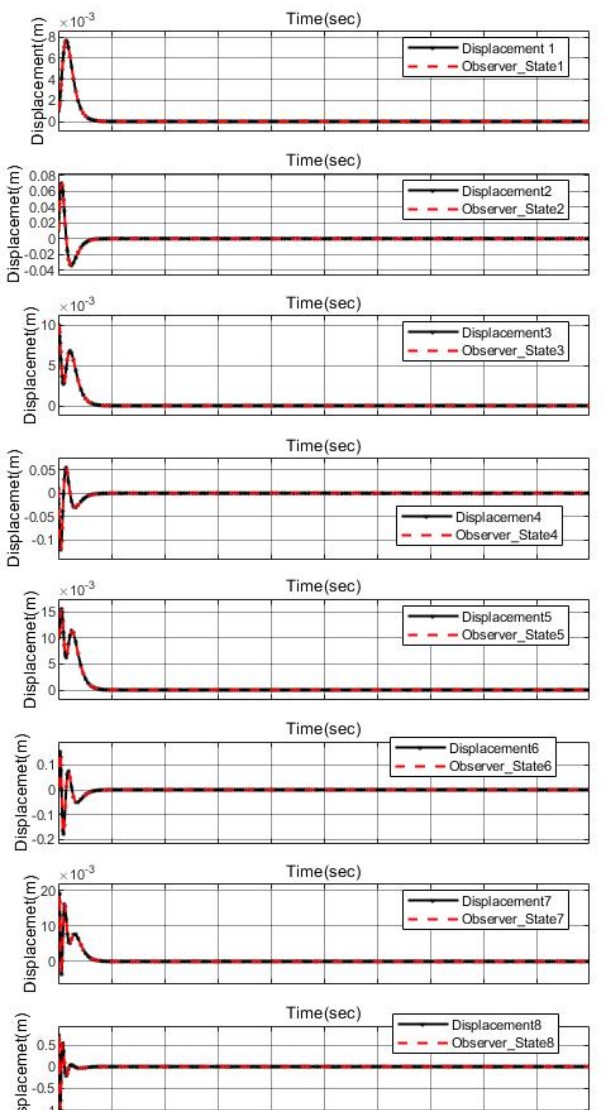

**Figure 15.** Controlled System States and Observer Signals for 8 Modes.

## 6. Conclusions

Our intent was to reduce microphonic interference in a nine-cell radio frequency cavity that has boundary conditions allowing access to only the ends of the cavity. As it is not possible to find an exact analytical model of a uniform nine-cell cavity, we modeled the cavity's dynamic as a cylindrical shell with the same thickness and length as the cavity, and we developed an analytical mechanical model for this structure with the same physical characteristics and eigenfrequencies. Utilizing a modified Fourier-Ritz method to fulfill the modelling, we presented a unified solution for a cylindrical shell system with general boundary conditions. Regardless of the boundary and continuity conditions, we were able to express each displacement of the cylindrical shell, using a modified Fourier series composed of the standard Fourier series and auxiliary functions. We determined all coefficients by the Rayleigh-Ritz method. Using ANSYS simulation, we conducted a modal analysis to compare the cavity's dynamic and the cylindrical shell's dynamic. The results showed that the first ten eigenfrequencies of the two structures are almost identical. Finally, we designed an active vibration control to cancel out specific mechanical modes of a nine-cell cavity with constrained access limiting measurement and application

of input forces to only the ends of the cavity. This observer-based controller can control a maximum of 8 modes of vibration. Using MATLAB simulation, we proved the accuracy of our observer-based LQG controller's ability to control all controllable modes of vibration.

**Author Contributions:** Conceptualization, M.K. and M.M.; methodology, M.K., M.M. and J.T.K.; software, M.K. and J.T.K.; validation, M.K., J.T.K. and M.M.; formal analysis, M.K., J.T.K. and M.M.; investigation, M.K. and M.M.; resources, M.M.; writing, M.K. and J.T.K.; writing—review and editing, M.M. and J.T.K; visualization, M.K.; supervision, M.M. and J.T.K.; project administration, M.M. All authors have read and agreed to the published version of the manuscript.

**Funding:** This research was funded by Simon Fraser University and partially by TRIUMF Canada Particle Accelerator Centre.

**Informed Consent Statement:** Not applicable.

**Data Availability Statement:** Not applicable.

**Conflicts of Interest:** The authors declare no conflict of interest.

## Appendix A. Displacement Functions in u, v and w Directions

$$u(x,\theta,r,t) = U(r,\theta)\cos(\lambda x)e^{j\omega t} \tag{A1}$$

$$v(x,\theta,r,t) = V(r,\theta)\sin(\lambda x)e^{j\omega t} \tag{A2}$$

$$w(x,\theta,r,t) = W(r,\theta)\cos(\lambda x)e^{j\omega t} \tag{A3}$$

where $\lambda = \frac{m\pi}{L}$, and

$$U(r,\theta) = C\Phi \tag{A4}$$

$$V(r,\theta) = \frac{1}{r}\frac{\Phi}{\theta} - \frac{\partial \Psi}{\partial r} \tag{A5}$$

$$W(r,\theta) = \frac{1}{r}\frac{\Phi}{\partial r} + \frac{1}{r}\frac{\partial \Psi}{\partial \theta} \tag{A6}$$

$$u(x,\theta,t) = \sum_{n=0}^{\infty}\sum_{m=0}^{\infty} u_{nm}(x,\theta,t) = \sum_{n=0}^{\infty}\sum_{m=0}^{\infty} U_{nm}\cos(\lambda_m x)\cos(n\theta)e^{j\omega t} \tag{A7}$$

$$v(x,\theta,t) = \sum_{n=0}^{\infty}\sum_{m=0}^{\infty} v_{nm}(x,\theta,t) = \sum_{n=0}^{\infty}\sum_{m=0}^{\infty} V_{nm}\sin(\lambda_m x)\sin(n\theta)e^{j\omega t} \tag{A8}$$

$$w(x,\theta,t) = \sum_{n=0}^{\infty}\sum_{m=0}^{\infty} w_{nm}(x,\theta,t) = \sum_{n=0}^{\infty}\sum_{m=0}^{\infty} W_{nm}\sin(\lambda_m x)\cos(n\theta)e^{j\omega t}. \tag{A9}$$

## Appendix B. Kinetic Energy and Potential Energy

$$E_{Kinetic} = \frac{\rho h}{2}\int_0^{2\pi}\int_0^L \left(\frac{\partial u}{\partial t}\right)^2 + \left(\frac{\partial v}{\partial t}\right)^2 + \left(\frac{\partial w}{\partial t}\right)^2 \times R\,dx\,d\theta \tag{A10}$$

where $\rho$ is the mass density of the cylindrical shell. Equations for strain in the cylindrical shell are

$$\epsilon_x = \frac{\partial u}{\partial x} - z\frac{\partial^2 w}{\partial x^2} \tag{A11}$$

$$\epsilon_\theta = \frac{1}{R}\frac{\partial v}{\partial \theta} + \frac{w}{R} - \frac{z}{R^2}\frac{\partial^2 w}{\partial \theta^2} \tag{A12}$$

$$\epsilon_{x\theta} = \frac{\partial v}{\partial x} + \frac{1}{R}\frac{\partial u}{\partial \theta} + \frac{2z}{R}\frac{\partial^2 w}{\partial x\partial \theta} \tag{A13}$$

$$\epsilon_{xz} = \epsilon_{\theta z} = \epsilon_{zz} = 0 \tag{A14}$$

and the equations for stress which are necessary to obtain the potential energy are:

$$\sigma_x = \frac{E}{1 - \mu^2}(\epsilon_x + \nu\epsilon_\theta) \tag{A15}$$

$$\sigma_\theta = \frac{E}{1 - \mu^2}(\epsilon_\theta + \mu\epsilon_x) \tag{A16}$$

$$\sigma_{x\theta} = \sigma_{\theta x} = \frac{E}{1 - \mu^2}\epsilon_{x\theta} \tag{A17}$$

$$\sigma_{xz} = \sigma_{\theta z} = \sigma_{zz} = 0 \tag{A18}$$

where E is Young's modulus and μ is the Poisson's ratio. So the potential energy can be expresses as

$$E_{Potential} = \frac{1}{2}\int_0^L \int_0^{2\pi} \int_0^h (\epsilon_x\sigma_x + \epsilon_\theta\sigma_\theta + \epsilon_{x\theta}\sigma_{x\theta})Rdxd\theta dz \tag{A19}$$

by inserting strain and stress into the potential energy equation.

$$
\begin{aligned}
E_{Potential} = \\
\frac{Eh}{2(1-\mu^2)}\int_0^{2\pi}\int_0^L &\left\{ \left(\frac{\partial u}{\partial x} + \frac{\partial v}{R\partial\theta} + \frac{w}{R}\right)^2 \right.\\
&- 2(1-\mu)\frac{\partial u}{\partial x}\left(\frac{\partial v}{R\partial\theta} + \frac{w}{R}\right)\\
&+ \left.\frac{(1-\mu)}{2}\left(\frac{\partial v}{\partial x} + \frac{\partial u}{r\partial\theta}\right)^2 \right\}Rdxd\theta\\
+ \frac{Eh^3}{24(1-\mu^2)}\int_0^{2\pi}\int_0^L &\left\{ \left(\frac{\partial^2 w}{\partial x^2} + \frac{\partial^2 w}{R^2\partial\theta^2}\right)^2 \right.\\
&- \left. 2(1-\mu)\frac{\partial^2 w}{\partial x^2}\frac{\partial^2 w}{R^2\partial\theta^2} - \left(\frac{\partial^2 w}{\partial x\partial\theta}\right)^2 \right\}Rdxd\theta\\
+ \frac{Eh^3}{24R^2(1-\mu^2)}\int_0^{2\pi}\int_0^L &\left\{ -2\mu\frac{\partial v}{\partial\theta}\frac{\partial^2 w}{\partial x^2} - 2\frac{\partial v}{\partial\theta}\frac{\partial^2 w}{R^2\partial\theta^2} \right.\\
&+ \left. \left(\frac{\partial v}{R\partial\theta}\right)^2 - 4(1-\mu)\frac{\partial v}{\partial x}\frac{\partial^2 w}{\partial x\partial\theta} + 2(1-\mu)\left(\frac{\partial v}{\partial x}\right)^2 \right\}
\end{aligned}
$$
$$Rdxd\theta \tag{A20}$$

**Appendix C. Dimension of Stiffness Matrix and Mass Matrix in Different Direction**

$$
[K_{uu}] = \begin{bmatrix}
[K_{uu11}^a]_{(MN\times MN)} & [K_{uu12}^a]_{(MN\times PN)} & \cdots \\
[K_{uu21}^a]_{(PN\times MN)} & [K_{uu22}^a]_{(PN\times PN)} & \cdots \\
0 & 0 & \cdots \\
0 & 0 & \cdots \\
0 & 0 & \\
0 & 0 & \\
[K_{uu11}^b]_{(MN\times MN)} & [K_{uu12}^b]_{(MN\times PN)} & \\
[K_{uu21}^b]_{(PN\times MN)} & [K_{uu22}^b]_{(PN\times PN)} &
\end{bmatrix} \tag{A21}
$$

where the matrix dimension is $(2N(M+P) \times 2N(M+P))$.

$$
[K_{uv}] = \begin{bmatrix}
0 & 0 & \cdots \\
0 & 0 & \cdots \\
[K^b_{uv11}]_{(MN \times MN)} & [K^b_{uv12}]_{(MN \times PN)} & \cdots \\
[K^b_{uv21}]_{(PN \times MN)} & [K^b_{uv22}]_{(PN \times PN)} & \cdots \\
[K^a_{uv11}]_{(MN \times MN)} & [K^a_{uv12}]_{(MN \times PN)} \\
[K^a_{uv21}]_{(PN \times MN)} & [K^a_{uv22}]_{(PN \times PN)} \\
0 & 0 \\
0 & 0
\end{bmatrix}
\tag{A22}
$$

where the matrix dimension is $(2N(M+P) \times 2N(M+P))$.

$$
[K_{uw}] = \begin{bmatrix}
[K^a_{uw11}]_{(MN \times MN)} & [K^a_{uw12}]_{(MN \times PN)} & \cdots \\
[K^a_{uw21}]_{(FN \times MN)} & [K^a_{uw22}]_{(FN \times PN)} & \cdots \\
0 & 0 & \cdots \\
0 & 0 & \cdots \\
0 & 0 \\
0 & 0 \\
[K^b_{uw11}]_{(MN \times MN)} & [K^b_{uw12}]_{(MN \times PN)} \\
[K^b_{uw21}]_{(FN \times MN)} & [K^b_{uw22}]_{(FN \times PN)}
\end{bmatrix}
\tag{A23}
$$

where the matrix dimension is $(2N(M+P) \times 2N(M+F))$.

$$
[K_{vv}] = \begin{bmatrix}
[K^a_{vv11}]_{(MN \times MN)} & [K^a_{vv12}]_{(MN \times PN)} & \cdots \\
[K^a_{vv21}]_{(PN \times MN)} & [K^a_{vv22}]_{(PN \times PN)} & \cdots \\
0 & 0 & \cdots \\
0 & 0 & \cdots \\
0 & 0 \\
0 & 0 \\
[K^b_{vv11}]_{(MN \times MN)} & [K^b_{vv12}]_{(MN \times PN)} \\
[K^b_{vv21}]_{(PN \times MN)} & [K^b_{vv22}]_{(PN \times PN)}
\end{bmatrix}
\tag{A24}
$$

where the matrix dimension is $(2N(M+P) \times 2N(M+P))$.

$$
[K_{vw}] = \begin{bmatrix}
0 & 0 & \cdots \\
0 & 0 & \cdots \\
[K^b_{vw11}]_{(MN \times MN)} & [K^b_{vw12}]_{(MN \times PN)} & \cdots \\
[K^b_{vw21}]_{(PN \times MN)} & [K^b_{vw22}]_{(PN \times FN)} & \cdots \\
[K^a_{vw11}]_{(MN \times MN)} & [K^a_{vw12}]_{(MN \times PN)} \\
[K^a_{vw21}]_{(PN \times MN)} & [K^a_{vw22}]_{(PN \times FN)} \\
0 & 0 \\
0 & 0
\end{bmatrix}
\tag{A25}
$$

where the matrix dimension is $(2N(M + P) \times 2N(M + F))$.

$$[K_{ww}] = \begin{bmatrix} [K^a_{ww11}]_{(MN \times MN)} & [K^a_{ww12}]_{(MN \times FN)} & \cdots \\ [K^a_{ww21}]_{(FN \times MN)} & [K^a_{ww22}]_{(FN \times FN)} & \cdots \\ 0 & 0 & \cdots \\ 0 & 0 & \cdots \\ 0 & 0 \\ 0 & 0 \\ [K^b_{ww11}]_{(MN \times MN)} & [K^b_{ww12}]_{(MN \times FN)} \\ [K^b_{ww21}]_{(FN \times MN)} & [K^b_{ww22}]_{(FN \times FN)} \end{bmatrix} \tag{A26}$$

where the matrix dimension is $(2N(M + F) \times 2N(M + F))$.

$$[M_{uu}] = \begin{bmatrix} [M^a_{uu11}]_{(MN \times MN)} & [M^a_{uu12}]_{(MN \times PN)} & \cdots \\ [M^a_{uu21}]_{(PN \times MN)} & [M^a_{uu22}]_{(PN \times PN)} & \cdots \\ 0 & 0 & \cdots \\ 0 & 0 & \cdots \\ 0 & 0 \\ 0 & 0 \\ [M^b_{uu11}]_{(MN \times MN)} & [M^b_{uu12}]_{(MN \times PN)} \\ [M^b_{uu21}]_{(PN \times MN)} & [M^b_{uu22}]_{(PN \times PN)} \end{bmatrix} \tag{A27}$$

where the matrix dimension is $(2N(M + P) \times 2N(M + P))$.

$$[M_{vv}] = \begin{bmatrix} [M^a_{vv11}]_{(MN \times MN)} & [M^a_{vv12}]_{(MN \times PN)} & \cdots \\ [M^a_{vv21}]_{(PN \times MN)} & [M^a_{vv22}]_{(PN \times PN)} & \cdots \\ 0 & 0 & \cdots \\ 0 & 0 & \cdots \\ 0 & 0 \\ 0 & 0 \\ [M^b_{vv11}]_{(MN \times MN)} & [M^b_{vv12}]_{(MN \times PN)} \\ [M^b_{vv21}]_{(PN \times MN)} & [M^b_{vv22}]_{(PN \times PN)} \end{bmatrix} \tag{A28}$$

where the matrix dimension is $(2N(M + P) \times 2N(M + P))$.

$$[M_{ww}] = \begin{bmatrix} [M^a_{ww11}]_{(MN \times MN)} & [M^a_{ww12}]_{(MN \times FN)} & \cdots \\ [M^a_{ww21}]_{(FN \times MN)} & [M^a_{ww22}]_{(FN \times FN)} & \cdots \\ 0 & 0 & \cdots \\ 0 & 0 & \cdots \\ 0 & 0 \\ 0 & 0 \\ [M^b_{ww11}]_{(MN \times MN)} & [M^b_{ww12}]_{(MN \times FN)} \\ [M^b_{ww21}]_{(FN \times MN)} & [M^b_{ww22}]_{(FN \times FN)} \end{bmatrix} \tag{A29}$$

where the matrix dimension is $(2N(M + F) \times 2N(M + F))$.

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
