# Peer review of "Radio Frequency Cavity’s Analytical Model and Control Design"

_vibration, doi:10.3390/vibration6020020_

Round 1
Reviewer 1 Report
Dear Authors:
The paper reports on an interesting topic enclosing a complete mathematical model as well as ANSYS validation. However, the experimental results are pre-collected so it is hard to understand the effectiveness of the proposed model in real-time measurement. In addition to the real-time experiment, please consider the below comments for revision.
1) Would you like to specify the significant contribution of this work? Is it a new model?
2) Art quality of Fig. 4-7 is very poor. It was difficult to follow the expression and Fig. 4-5. Please update them.
3) Please specify the type of piezo sensor that you used. Was it a microphone, force sensor, or what?
4) Please update Fig. 13. Very poor image quality.
5) It is recommended to show the pressure distribution inside the cavity.
6) Is there any optimization and/or study behind the nine-cell cavity selection?
Author Response
- Would you like to specify the significant contribution of this work? Is it a new model?
Thank you so much for your review. To discuss more about the significant contribution of our work, we can mention that different studies on suppression of mechanical vibration have been conducted in accelerator labs all around the world; these have shown that feedback and feed-forward methods are necessary to suppress these unknown and unwanted mechanical vibrations. While various computer programs have been used to calculate the resonant frequency and field strength (electric and magnetic) of the modes of interest (based on Maxwell’s Equations and the boundary conditions), analytic solutions for RF fields in an RF structure are not available except for simple geometries. Also, an analytical model of multi-cell cavity for the mechanical vibration has not been available. In this research we focused on the analytical model of the nine-cell cavity which has been missing in other studies. That’s our contribution in introducing of this new model.
- Art quality of Fig. 4-7 is very poor. It was difficult to follow the expression and Fig. 4-5. Please update them.
Thank you so much for the feedback. we updated the figures.
- Please specify the type of piezo sensor that you used. Was it a microphone, force sensor, or what?
In our studies we placed stack piezo at the ends of two coupled cavities and measured the vibration without and with the piezo. This piezo stack has been measured the length change of the cavity under vibration.
- Please update Fig. 13. Very poor image quality.
Thank you so much for the feedback. We updated the figure.
- It is recommended to show the pressure distribution inside the cavity.
Thank you for your valuable comment. We completely agree with you and took your suggestion into consideration. As a result, we have reviewed the possibility of showing the pressure inside the cavity but as this study is a simulation and we didn’t have access to measure the Pressure distribution inside the actual cavity. Due to many restrictions about doing a real measurement on the cavity we just limited our studies to the simulation to electromagnetic field distributions. For further studies it’s suggested that research with all measurement on the cavity be conducted.
- Is there any optimization and/or study behind the nine-cell cavity selection?
This nine-cell cavity has been studied in many researches in all particle accelerator around the world specially at CERN. Initially this nine-cell cavity has been introduced under TESLA project in DASY lab, Germany. Considering the dimension, scientists came to this conclusion that the best structure to accelerate the beam of particles would be this elliptical nine-cell superconducting cavity. However other types of cavities are also has been utilized to accelerate the beam of particles. More information about the history of this nine-cell cavity can be found in CERN website. (https://cerncourier.com/a/teslas-high-gradient-march/)
Reviewer 2 Report
The authors carry out analytical studies on the modal characteristics of a cylindrical shell in aims of reflecting the dynamics of a more complicated structure of a nine-cell cavity shells. Then they propose an active vibration control scheme for the shell structure. I have the following concerns and questions before I can recommend it to be published.
- I ‘m not fully convinced of the general idea of using a simple cylindrical shell to represent the modal properties of the multi-cavity RF structure. From Fig. 6 and 7 , their modal frequencies seem to have a very large difference . For higher modes, neither the modal frequencies nor the mode shapes are comparable. If this is the case, the future prospects of this work is not very promising. The authors need to further explain the correspondence between the simplified model and the real structure.
- How the equivalent length of the shell is calculated?
- The vibration of the cylindrical shell is well studied , as well as the active vibration control methods of the shell. What is the novelty of this work? The authors need to explicitly explain this in the manuscript.
- How the will the control system be used for the real RF structure ?
Author Response
- I ‘m not fully convinced of the general idea of using a simple cylindrical shell to represent the modal properties of the multi-cavity RF structure. From Fig. 6 and 7, their modal frequencies seem to have a very large difference. For higher modes, neither the modal frequencies nor the mode shapes are comparable. If this is the case, the future prospects of this work is not very promising. The authors need to further explain the correspondence between the simplified model and the real structure.
We very much appreciate this helpful comments. To address this feedback we have to mention that to develop a controller to decrease or suppress microphonic interference, an adequate model of such a cavity is required first. Historically, developing an analytical model of mechanical vibrations in a multi-cell cavity has proven to be an incredibly difficult undertaking. Further challenges in controlling TRIUMF’s nine-cell niobium RF cavity arise from its boundary conditions. Access to the cavity is restricted to either end because the cavity is suspended within a Helium bath, limiting applications of forces only to the cavity ends. The vibration which are causing problem in our TRIUM’s nine-cell cavity that we studied and find them through our measurement are the frequencies that we suppressed through our proposed control design and other vibration does not have that much effect on the beam of the particles in our particular case.
- How is the equivalent length of the shell calculated?
The equivalent length of the shell calculated through finding the similar natural frequency with the nine-cell cavity. We first find the natural frequencies of the nine-cell cavity through ANSYS and then find another cylindrical model with exactly the same natural frequencies.
- The vibration of the cylindrical shell is well studied, as well as the active vibration control methods of the shell. What is the novelty of this work? The authors need to explicitly explain this in the manuscript.
We modeled the multi-cell cavity’s dynamic as a cylindrical shell, which has a mathematical model that is ideal for replacing the nine-cell cavity in dynamic modelling. This resulted in a unified solution for cylindrical shell systems with generic boundary conditions. From this model, we developed an observer-based LQG controller—a combined Kalman filter and LQR controller—and proved its accuracy and effectiveness through simulation analysis.
- How the will the control system be used for the real RF structure?
Conducting further vibration measurement of the actual cavity at TRIUMF and running experimental studies with the selected piezo sensors and their integration into the control design would be an appropriate next step in this research. For example, application of a shaker to cause some vibrations and then using a phase noise analyzer that detects the cavity resonance frequency could be used for the measurement studies.
Round 2
Reviewer 2 Report
The authors have addressed most of my questions. Though I still have my concerns about the scientific soundness of this work, I think the authors have presented a complete piece of story that could be interesting to engineering fields.